# Bandwidth Optimization of MEMS Accelerometers in Fluid Medium Environment

**DOI:** 10.3390/s22249855

**Published:** 2022-12-15

**Authors:** Xiang Xu, Shuang Wu, Weidong Fang, Zhe Yu, Zeyu Jia, Xiaoxu Wang, Jian Bai, Qianbo Lu

**Affiliations:** 1Frontiers Science Center for Flexible Electronics (FSCFE), MIIT Key Laboratory of Flexible Electronics (KLoFE), Shaanxi Key Laboratory of Flexible Electronics (KLoFE), Institute of Flexible Electronics (IFE), Ningbo Institute of Northwestern Polytechnical University, Northwestern Polytechnical University, Xi’an 710072, China; 2State Key Laboratory of Modern Optical Instrumentation, Zhejiang University, Hangzhou 310027, China; 3The Key Laboratory of Information Fusion Technology, Ministry of Education, School of Automation, Northwestern Polytechnical University, 127 West Youyi Road, Xi’an 710072, China

**Keywords:** bandwidth optimization, MEMS accelerometer, squeeze film damping, amplitude effect

## Abstract

There is a constraint between the dynamic range and the bandwidth of MEMS accelerometers. When the input acceleration is comparatively large, the squeeze film damping will increase dramatically with the increase in the oscillation amplitude, resulting in a decrease in bandwidth. Conventional models still lack a complete vibration response analysis in large amplitude ratios and cannot offer a suitable guide in the optimization of such devices. In this paper, the vibration response analysis of the sensing unit of an accelerometer in large amplitude ratios is first completed. Then, the optimal design of the sensing unit is proposed to solve the contradiction between the dynamic range and the bandwidth of the accelerometer. Finally, the results of the vibration experiment prove that the maximum bandwidth can be achieved with 0~10*g* external acceleration, which shows the effectiveness of the design guide. The new vibration analysis with the complete model of squeeze film damping is applicable to all sensitive structures based on vibration, not limited to the MEMS accelerometer studied in this thesis. The bandwidth optimal scheme also provides a strong reference for similar structures with large oscillation amplitude ratios.

## 1. Introduction

Among various MEMS devices, especially those with micro-cantilever structures or composite piezoelectric materials [1,2,3,4,5], when the oscillation amplitude of the structure is comparable to the air film thickness, it can lead to an entirely different dynamic damping effect [6,7]. The vibration response of the structures and the dynamic performance of the devices are thus influenced. Therefore, understanding the vibration in large amplitude ratios carries great significance for the further optimal design of such MEMS devices.

Particularly, for MEMS accelerometers in a fluid medium environment, the squeeze film air damping will increase dramatically with the increase in oscillation amplitude caused by large external acceleration [8]. However, according to the design conventions, the optimal damping ratio should be 0.707 to obtain the maximum bandwidth [9]. When the damping ratio increases owing to the amplitude effect at large external acceleration, the bandwidth will decrease subsequently. In other words, the dynamic range of the accelerometer will be restricted while keeping the bandwidth constant. It is necessary to make trade-offs between them to obtain the most suitable parameter design.

Since the bandwidth of the accelerometer is closely related to the squeeze film damping of the structure, the bandwidth optimization of the accelerometer relies on the optimal design of squeeze film damping. Based on the solution equation of squeeze film damping [10], there are mainly two optimal schemes of damping in MEMS accelerometers: the shape optimization of the vibrating plate and the viscous coefficient control of the fluid medium. The shape optimization of the vibrating plate can be traced back to 1970. Li et al. [11] first proposed the optimal design of the capacitive accelerometer to obtain the ideal bandwidth based on the theory of squeeze film damping. Since then, the design of squeeze film damping based on various types of vibrating plates has been widely used to optimize the performance of accelerometers, such as noise [12,13,14] and bandwidth [15,16,17,18]. Another way is realized through the viscous coefficient control of the fluid medium. Rudolf [19] proposed a micromechanical capacitive accelerometer with a two-point inertia mass suspension. Through the tests of vibration response in atmospheric and vacuum environments, he first proved the possibility of changing the vibration response through the air pressure around the structure. Kavitha [20], Aono [21], and Mo et al. [22] also conducted similar studies on the viscous coefficient’s effect on the performance of micro-accelerometer. However, in all the above designs, the damping coefficients are considered constant in the vibration analysis, which only satisfies the small amplitude ratio situations.

In this paper, we first discuss the physical model of squeeze film damping and extend the vibration response analysis to the large amplitude ratio situations. Thereafter, we turn to the shape optimization of the sensing unit to accomplish the bandwidth optimization in our study combined with the vibration response analysis in a large amplitude ratio. Experiment measurements based on the half-power bandwidth method further confirm the effectiveness of our optimal bandwidth scheme. In addition, the whole optimal design is rather simple and practical, which can be considered a strong guide in the damping-related analysis and corresponding optimal MEMS design.

## 2. Vibration Response Analysis of a MEMS Accelerometer

The current dynamic response analysis of the vibrating system is mainly limited to the cases where the damping coefficient is treated as a constant, corresponding to the situations where the oscillation amplitude ratio (the ratio of the oscillation amplitude of the plate to the initial thickness of the air film) is small. Owing to the lack of systematic vibration response analysis in large oscillation amplitude ratio situations, where the damping coefficient is greatly affected by the amplitude effect, the related work has no choice but to rely on the existing model or finite element analysis, impeding further performance optimization of these devices.

In this section, we analyze the vibration response of the second-order damped vibration system in the attenuation of oscillation and forced vibration, respectively. First, we discuss the typical vibration response of the classical system where the damping coefficient is fixed. Then, we derive a complete model of vibration response where the squeeze film damping coefficient is influenced by the amplitude effect, which extends the validity of the origin vibration response analysis theory. It also paves the way for the analysis and optimization of the accelerometer in the following sections.

### 2.1. Attenuation of Oscillation in Large Amplitude Ratio

In 1962, Langlois [23] proposed the first-order approximation of the squeeze film damping force and divided the damping force into two parts: the spring damping force proportional to the displacement of the plate and the viscous damping force proportional to the velocity of the plate. This analysis introduced the squeeze film damping into a first-order spring damper system, thus providing the most important theoretical basis for combining the squeeze film damping with the physical model of the accelerometer.

For a damped vibration system, as shown in Figure 1, the differential equation for the movement is the second-order differential equation:(1)$$m\ddot{z}=-kz-F_{d}$$
where *m* is the mass of the proof mass, *k* is the elastic coefficient of the spring, *z* is the displacement in the sensitive axis direction, and *F_d_* is the damping force of the system.

For a MEMS accelerometer, the damping force mainly comes from the squeeze film damping caused by air between the mass and the glass cover (or substrate). Based on the complete squeeze film damping model, the value of the damping coefficient is affected by the oscillation amplitude. In the large amplitude ratio cases, the damping force can be written as follows [8]:(2)$$F_{d}=c_{d}\dot{z}\approx u\left(\varepsilon \right)c_{0}\dot{z}$$
where *ε* is the amplitude ratio, *c*_0_ is the damping coefficient in the small amplitude ratio, *c_d_* is the damping coefficient in large amplitude ratio, and
(3)$$c_{0}=f\left(\gamma \right){\left(1+\frac{8}{3\pi }\frac{h_{0}}{2a}\right)}^{4}\frac{\mu a^{4}}{h_{0}^{3}}$$
and
(4)$$u\left(\varepsilon \right)=\frac{\left[\begin{array}{l} \frac{1}{{\left(1-\varepsilon^{2}\right)}^{1.5}}+\left(3+\gamma \right)\frac{2\left(1-\sqrt{1-\varepsilon^{2}}\right)}{\varepsilon^{2}\sqrt{1-\varepsilon^{2}}}\beta \\ +6\left(1+\gamma \right)\frac{\left(1-\sqrt{1-\varepsilon^{2}}\right)}{\varepsilon^{2}}\beta^{2}+\left(3\gamma +1\right)\beta^{3}+\gamma \beta^{4} \end{array}\right]}{{\left(1+\beta \right)}^{4}}$$
where *γ* is the aspect ratio of the plate, *h*_0_ is the thickness of the air film, *a* is the characteristic length of the plate, and *μ* is the viscous coefficient. For the rectangular plate,
(5)$$f(\gamma )=16\gamma \left(1-\frac{192\gamma }{n^{5}}\sum_{n=\mathrm{odd}}\frac{1}{n^{5}}\tanh\frac{n\pi }{2\gamma }\right)$$

By substituting Equation (2) into Equation (1), we obtain
(6)$$\ddot{z}+\frac{c_{d}}{m}\dot{z}+\frac{k}{m}z=0$$

Using $\omega_{0}=\sqrt{\frac{k}{m}}$ and $\zeta =\frac{c_{0}}{2\sqrt{km}}$, we have the free decay second-order equation of the spring damping system as follows:(7)$$\ddot{z}+2\frac{c_{d}}{c_{0}}\zeta \omega_{0}\dot{z}+\omega_{0}^{2}z=0$$
where *ω*_0_ is the natural frequency of the vibration system if there is no damping and *ζ* is called the damping ratio; the main parameter indicates the whole system’s damping effect.

By letting $z=z_{0}e^{\delta t}$, we obtain the solution of the equation as
(8)$$z=\{\begin{array}{cc} z_{0}e^{-\frac{c_{d}}{c_{0}}\zeta \omega_{0}t}\sin\left(\omega_{0}\sqrt{1-\frac{c_{d}^{2}}{c_{0}^{2}}\zeta^{2}}t+\alpha \right) & \frac{c_{d}}{c_{0}}\zeta <1①\\ z_{0}e^{-\frac{c_{d}}{c_{0}}\zeta \omega_{0}t} & \frac{c_{d}}{c_{0}}\zeta =1②\\ z_{0}e^{-\frac{c_{d}}{c_{0}}\zeta \omega_{0}t}\left(c_{1}e^{\omega_{0}\sqrt{\frac{c_{d}^{2}}{c_{0}^{2}}\zeta^{2}-1}t}+c_{2}e^{-\omega_{0}\sqrt{\frac{c_{d}^{2}}{c_{0}^{2}}\zeta^{2}-1}t}\right) & \frac{c_{d}}{c_{0}}\zeta >1③ \end{array}$$

In Formula ① in Equation (8), the process of decay is accompanied by the simple harmonic vibration; thus
(9)$$\frac{c_{d}}{c_{0}}=u\left(\varepsilon \right)=\frac{\left[\begin{array}{l} \frac{1}{{\left(1-\varepsilon^{2}\right)}^{1.5}}+\left(3+\gamma \right)\frac{2\left(1-\sqrt{1-\varepsilon^{2}}\right)}{\varepsilon^{2}\sqrt{1-\varepsilon^{2}}}\beta \\ +6\left(1+\gamma \right)\frac{\left(1-\sqrt{1-\varepsilon^{2}}\right)}{\varepsilon^{2}}\beta^{2}+\left(3\gamma +1\right)\beta^{3}+\gamma \beta^{4} \end{array}\right]}{{\left(1+\beta \right)}^{4}}$$

From the definition of amplitude ratio, we obtain
(10)$$\varepsilon =\frac{z_{0}}{h_{0}}e^{-u\left(\varepsilon \right)\zeta \omega_{0}t}$$

It is clear that *u*(*ε*) is a monotonically increasing function of *ε* with a value range from 1 to *+∞*. Therefore, for a vibration system with a specific initial condition, there is only one solution *ε*, which satisfies Equation (10).

Regarding Formula ② and Formula ③ in Equation (8), the decay process is not accompanied by simple harmonic vibration, and in this case,
(11)$$\frac{c_{d}}{c_{0}}=\frac{f\left(\gamma \right){\left(1+\frac{8}{3\pi }\frac{h_{d}}{2a}\right)}^{4}\frac{\mu a^{4}}{h_{d}^{3}}}{f\left(\gamma \right){\left(1+\frac{8}{3\pi }\frac{h_{0}}{2a}\right)}^{4}\frac{\mu a^{4}}{h_{0}^{3}}}={\left(\frac{1+h_{d}\beta /h_{0}}{1+\beta }\right)}^{4}\frac{h_{0}^{3}}{h_{d}^{3}}$$
where *h_d_* is the transient thickness of the air film with value *h*_0_ ± *z* (the sign depends on the direction from the equilibrium position at the start-up of vibration). So, Formula ② and Formula ③ satisfy the equation sets
(12)$$\{\begin{array}{l} z=z_{0}e^{-\frac{c_{d}}{c_{0}}\zeta \omega_{0}t}\\ \frac{c_{d}}{c_{0}}={\left(\frac{1+\left(h_{0}\pm z\right)\beta /h_{0}}{1+\beta }\right)}^{4}\frac{h_{0}^{3}}{{\left(h_{0}\pm z\right)}^{3}} \end{array}$$
and
(13)$$\{\begin{array}{l} z=z_{0}e^{-\frac{c_{d}}{c_{0}}\zeta \omega_{0}t}\left(c_{1}e^{\omega_{0}\sqrt{\frac{c_{d}^{2}}{c_{0}^{2}}\zeta^{2}-1}t}+c_{2}e^{-\omega_{0}\sqrt{\frac{c_{d}^{2}}{c_{0}^{2}}\zeta^{2}-1}t}\right)\\ \frac{c_{d}}{c_{0}}={\left(\frac{1+\left(h_{0}\pm z\right)\beta /h_{0}}{1+\beta }\right)}^{4}\frac{h_{0}^{3}}{{\left(h_{0}\pm z\right)}^{3}} \end{array}$$
respectively.

In summary, when the position of seismic mass significantly deviates from the equilibrium position due to a large amplitude ratio, the free decay process differs from that in a small amplitude ratio situation. Constricted by the complexity of the equations, a simple form of an analytical solution cannot be obtained, but we can derive the numerical results in different damping ratios by solving Equations (10), (12), and (13).

### 2.2. Forced Vibration in Large Amplitude Ratio

In many cases, a mass–spring system is excited into vibration by an external force. For example, the accelerometer in our study is mounted on a vibrator in a piggy-back form. The inertial force applied to the seismic mass of the accelerometer is related to the frequency of the vibration and the oscillation amplitude.

With reference to the inertia coordinate system *Oz*, if the vibration of the machine is $z_{M}=d_{0}\sin\left(\omega t\right)$, where *d*_0_ is the amplitude of the vibration and *ω* is the vibration frequency. We term the displacement of the seismic mass as *z_m_* and substitute the related parameters into Equation (6) and obtain
(14)$${\ddot{z}}_{m}+\frac{c_{d}}{m}\left({\dot{z}}_{m}-{\dot{z}}_{M}\right)+\frac{k}{m}\left(z_{m}-z_{M}\right)=0$$

Taking the relative displacement of the seismic mass as *z*
*=*
*z_m_*− *z_M_*, and substituting it into the equation above, we obtain
(15)$$\ddot{z}+\frac{c_{d}}{m}\dot{z}+\frac{k}{m}z=-{\ddot{z}}_{M}=d_{0}\omega^{2}\sin\left(\omega t\right)$$

The solution to Equation (15) takes the form *z = z*_1_
*+ z*_2_, where *z*_1_ is a general solution to the homogenous differential equation of the damped system, and *z*_2_ is a particular solution. Referring to the solution in a small amplitude situation, we know that the general solution indicates that transient oscillation occurs due to forced vibration with damping, which is usually short-lived and thus can be ignored in most cases where the damping coefficient is large enough. However, the particular solution is the actual vibration output when conditions are steady and determines the sensitivity and bandwidth of the accelerometer, which we really care about. It satisfies
(16)$${\ddot{z}}_{2}+\frac{c_{d}}{m}{\dot{z}}_{2}+\frac{k}{m}z_{2}=d_{0}\omega^{2}\sin\left(\omega t\right)$$

Similarly, by substituting the natural frequency *ω*_0_, damping ratio *ζ*, and the particular solution $z_{2}=M_{2}\sin\left(\omega t-\phi \right)$ in the above equation, we obtain
(17)$$\begin{array}{l} \left[M_{2}\left(\omega_{0}^{2}-\omega^{2}\right)-d_{0}\omega^{2}\cos\phi \right]\sin\left(\omega t-\phi \right)\\ +\left(2M_{2}u\left(\frac{M_{2}}{h_{0}}\right)\zeta \omega_{0}\omega -d_{0}\omega^{2}\sin\phi \right)\cos\left(\omega t-\phi \right)=0 \end{array}$$

Because the above equation is constantly equal to zero, we can obtain the equation set as follows:(18)$$\{\begin{array}{l} M_{2}^{2}{\left(\omega_{0}^{2}-\omega^{2}\right)}^{2}+4M_{2}^{2}u^{2}\left(\frac{M_{2}}{h_{0}}\right)\zeta^{2}\omega_{0}^{2}\omega^{2}=d_{0}^{2}\omega^{4}\\ \phi =\arctan\left(\frac{2u\left(\frac{M_{2}}{h_{0}}\right)\zeta \omega_{0}\omega }{\omega_{0}^{2}-\omega^{2}}\right) \end{array}$$

Similarly, there must be only one real solution to *M_ω_* and *ϕ_ω_* in the above equation. Then, the steady vibration response is obtained as
(19)$$z_{2}=M_{\omega }\sin\left(\omega t-\phi_{\omega }\right)$$

Although we cannot derive the exact analytical solution from the above equation set, the normalized amplitude response curve can be obtained by numerical methods.

From the discussion above, we first obtain the numerical solution of vibration response in large amplitude ratio cases based on the complete model of squeeze film damping, which paves the way for the bandwidth optimization of the accelerometer in the next section.

## 3. Accelerometer Bandwidth Optimization Based on the Complete Model of Squeeze Film Damping and Vibration Response Analysis

As can be seen in Figure 2, for a typical vibration system, the theoretical damping ratio of the structure is often designed to be 0.707 to obtain the maximum operating bandwidth according to an error standard of 3 dB. However, from the discussion in Section 2, it is obvious that the damping ratio will increase by more than four times considering the amplitude effect, resulting in a significant decrease in the bandwidth of the system.

Because in most non-vacuum-packaged MEMS devices, the fluid medium is generally air, the optimal design scheme proposed in this section also uses air as the fluid medium, and its viscous coefficient is 1.7984 × 10^−5^ N·s/m^2^. In our study, the sensing unit of the accelerometer is a sandwiched proof mass with air film on both sides, which is shown in Figure 3. Both the proof mass and the cantilever beam of the sensing unit are made from silicon, with a density of 2330 kg/m^3^ and an elastic modulus of 166 Gpa.

The proof mass is affected by the air-damping force on both sides in opposite directions, so the second-order differential equation of the vibrating system can be written as
(20)$$m\ddot{z}=-kz-F_{d1}+F_{d2}$$
where *F_d_*_1_ and *F_d_*_2_ are the squeeze film damping forces on each side, respectively. For a two-side symmetric structure, the damping coefficients on each side are equal during the vibration process, and the phase difference of the displacement function relative to the air film of each side is *π*.
(21)$$\{\begin{array}{l} F_{d1}=c_{d}\omega \cos\left(\omega t-\phi \right)\\ F_{d2}=c_{d}\omega \cos\left(\omega t-\phi -\pi \right)=-F_{d1} \end{array}$$

Thus, the second-order differential equation above can be simplified as below:(22)$$m\ddot{z}=-kz-2F_{d1}=-kz-2c_{d}\dot{z}$$

It is obvious that the squeeze film damping in the double-sided symmetrical sandwich structure is twice the value in the single-sided structure. For the convenience of description, the squeeze film damping coefficient refers to the sum of both sides in the article.

The vibration responses of the open-loop accelerometer have two cases. When the input acceleration is small, and the oscillation amplitude of the proof mass is not large enough to cause the amplitude effect, the actual vibration response satisfies the original theory with a fixed damping ratio. However, when the input acceleration is large enough, the amplitude of the proof mass will grow several times compared with the first case. Moreover, the ratio of the oscillation amplitude to the air film thickness determines whether it satisfies the small or large amplitude ratio case, which we briefly discuss as follows.

Since the elastic coefficient of the cantilever beam has no substantial influence on the optimal bandwidth scheme, for the convenience of manufacturing, we use the straight beam in our sensing unit to support the proof mass. Our previous research [24] has shown that our sandwiched accelerometer can have a maximum input acceleration of ±10*g*, with a displacement–acceleration sensitivity of 8.05 μm/g and a natural frequency of 175.5 Hz. With the maximum acceleration input, the displacement of the proof mass can reach up to 80.5 μm. According to the coefficient *u*(*ε*, *β*, *γ*) introduced by the amplitude effect in Equation (9), the thickness of air film should be more than five times the maximum amplitude; then, the amplitude effect can be ignored (*ε* = 0.2, the error reaches 6%). From the analysis in Section 2, the damping ratio should be 0.707 in the ideal situation to obtain the maximum bandwidth of the accelerometer. Then, by using the formula of the damping ratio,
(23)$$\zeta =\frac{c_{0}}{2\sqrt{km}}=\frac{0.42\mu {\left(2a\right)}^{3}{\left(1+\beta \right)}^{4}}{h_{0}^{3}\sqrt{k\rho t_{p}}}$$
we can obtain the thickness of the air film as 400 μm and the length of the proof mass as 26 mm, which is eight times larger than the size before optimization. The size is far beyond the acceptable range of our structure size. Therefore, for the high-sensitivity accelerometer studied in this paper under appropriate size, the dynamic response of a large acceleration load must follow the analysis results in Section 2.

For convenience, we called the first case (*ε* = 0) the base state and the second case (*ε* = 0.8) the large-amplitude state, and the corresponding damping ratios are called the basic damping ratio and the large-amplitude damping ratio, respectively.

The micro-nano fabrication technology generally uses a single crystal silicon wafer or SOI wafer as the main material, whose thickness is usually between 100 μm and 500 μm. At the same time, it is considered that in the deep etching process, the deeper the etching depth, the worse the structural flatness and parallelism. Therefore, the final air film thickness and the proof mass thickness are chosen as 100 μm and 400 μm, respectively. Under the target air film thickness, the basic damping ratio of the structure now is only 0.074, far smaller than the ideal damping ratio of 0.707. Figure 4 shows the normalized amplitude response curve when the basic damping ratio is 0.074 in the basic state. Referring to the 3 dB standard, the theoretical bandwidth is only 50% of its natural frequency. When the vibration frequency is close to the natural frequency, there will be a huge peak, which should be avoided for non-resonant accelerometers.

According to the formula of amplitude effect, it is obvious that the damping ratio will increase with the increase in oscillation amplitude. Then, we can divide the situation into three cases.

I. In the basic state where the amplitude ratio *ε* ≈ 0, the basic damping ratio *ζ*_0_ = 0.707.

In this case, when the input acceleration increases from 1*g* to 10*g*, the damping ratio will increase from 0.707 to 3.08. From the normalized amplitude curve shown in Figure 5, we know the bandwidth is only 88 Hz.

II. In the large-amplitude state, where the amplitude ratio *ε* ≈ 0.8, the large-amplitude damping ratio *ζ_ε_* = 0.707.

In this case, with the system damping ratio being 0.707 at the maximum input acceleration, the damping ratio varies from 0.162 to 0.707 in the input acceleration range. As shown in Figure 6, the normalized amplitude curve indicates that the bandwidth of the accelerometer is 98 Hz.

III. In the basic state, the damping ratio *ζ*_0_ < 0.707, and in the large-amplitude state, the damping ratio *ζ_ε_* > 0.707.

Obviously, the above schemes cannot meet our requirements for bandwidth. Although they can obtain a suitable amplitude frequency response, the whole range of accelerometer bandwidth is suppressed. From the discussion above, we can know that the key to bandwidth optimization is choosing a reasonable variable range of damping ratio, and then the whole optimization process can be described as in the flowchart shown in Figure 7.

In the basic state, using the 3 dB standard and by defining the ratio of bandwidth to the natural frequency as the normalized bandwidth coefficient, we can derive the curve of the normalized bandwidth coefficient as a function of the system damping ratio, as shown in Figure 8. The blue coverage area in the figure represents the area where the normalized bandwidth coefficient is greater than 1. It can be inferred that when the basic damping ratio is within the range of 0.384~0.707, the normalized bandwidth coefficient always remains greater than 1, which means the basic state in this area can improve the bandwidth.

Then, by substituting the basic damping ratio *ζ*_0_ into Equation (23), we can obtain the curve of characteristic length of proof mass with different basic damping ratios, as shown in Figure 9.

Substituting the derived proof mass length into the complete squeeze film damping model, considering the amplitude effect, we can obtain the large-amplitude damping ratio *ζ_ε_* and the normalized bandwidth coefficient with the corresponding basic damping ratio *ζ*_0_ demonstrated in Figure 10. Similarly, the blue coverage area is where the normalized bandwidth coefficient is larger than 1, corresponding to the area where the basic damping ratio *ζ*_0_ < 0.401. Combining with the range of damping ratio state shown in Figure 8, the basic damping ratio should be chosen from 0.384 to 0.401 to achieve the maximum accelerometer bandwidth.

The basic damping ratio is finally determined to be 0.39, considering the micromachining accuracy. At the same time, the natural frequency of the structure remains the same. Then, by solving the equation for basic damping ratio ζ, natural frequency ω_0_, and mass m together, we can obtain the value of the characteristic length of the proof mass and the elastic coefficient of the cantilever beam. They are the final parameters of the optimal structure. The parameters of the initial structure and the optimal structure are listed in Table 1.

## 4. Vibration Experiment of the Sandwiched Accelerometer

In this section, vibration experiments are performed on the sensing unit of the accelerometer to obtain its frequency response curve to verify the effectiveness of bandwidth optimization.

The structures of the sensing units used in our experiments are shown in Figure 11, which are symmetrical and sandwiched, and the parameters are listed in Table 1. Consistent with Figure 3, the structures consist of three layers, namely substrate, sensitive structure, and cover from the bottom to the top, whose materials are silicon, silicon, and glass. There is a groove in both the substrate and the cover to leave enough room for the displacement of the proof mass, whose thickness is also the air film thickness.

The experimental equipment is shown in Figure 12. The vibration responses under different frequencies and accelerations were measured in the experiment to verify the effectiveness of optimal bandwidth design. The structure of the sensing unit was glued on the surface of a vibrator with its substrate in contact with the vibrator. The signal generator (RIFOL DG1011) and the power amplifier (Labworks Inc. pa-138) were used to apply equivalent inertial acceleration at different vibration frequencies to the sensing unit. In order to reduce the measuring error, a rangefinder (RIFTEK RF603) was selected to measure the displacement of the shaker surface directly. A Doppler Vibrometer (Polytec PDV 100) was chosen to measure the absolute displacement of the sensing unit. The signal was proven to be not affected by the light reflected from the upper glass layer and can obtain accurate information on the proof mass vibration. Finally, the signal test and analysis system (DONGHYATEST DH5927N) collected the output signals of the rangefinder and vibrometer, and transmitted them to the host computer. After processing the above two signals, we obtained the relative displacement of proof mass to the vibrator surface against time.

According to the formula of the equivalent inertia acceleration *a* = *d*_0_(2*πf*)^2^, we can apply different accelerations to the sensing unit by controlling the amplitude *d*_0_ and frequency *f* through the signal generator and power amplifier. In the experiment, 1*g*, 5*g*, and 10*g* equivalent inertia accelerations were applied to the sensing unit, and the vibration frequency range is from 40 Hz to 200 Hz with an interval of 20 Hz.

Figure 13 demonstrates the absolute displacement curve of the proof mass and vibrator against time, respectively, when the equivalent acceleration is 5*g* and the vibration frequency is 100 Hz. The dashed lines in red and blue in Figure 13 represent the sine fitting curves of the vibrator displacement and the mass displacement, respectively. The R^2^ of the two fitting curves are 0.999 and 0.983, respectively, indicating that the output signals are in agreement with the theory.

The curves of normalized amplitude under different frequencies in 1*g*, 5*g*, and 10*g* equivalent inertial accelerations input are demonstrated in Figure 14. The blue coverage area represents the frequency range within the standard error of 3 dB for bandwidth. Because the vibration amplitude of the structure is normalized, the upper and lower limits of the normalized amplitude within the bandwidth are 1.414 and 0.707, respectively. As shown in Figure 14a, when the input equivalent inertial acceleration is 1*g*, the normalized amplitude of the initial structure is beyond the blue coverage area in most of the frequency range, and the maximum amplitude can reach up to three times the ideal amplitude. With the increase in acceleration, the normalized amplitude gradually meets the deviation requirement of bandwidth. When the input equivalent acceleration is 10*g*, the blue area covers the whole range of frequency, no less than that of the optimal structure. However, unlike the initial structure, the normalized amplitude of the optimal structure consistently meets the requirement deviation of bandwidth, which means its actual bandwidth can be considered to have reached the target of 175.5 Hz.

In the comparison of the normalized amplitude response at different frequencies of the initial structure and optimal structure, it is obvious that the overall frequency response curve flatness of the optimal structure is significantly better than that of the initial structure. This proves the effectiveness of the optimal design, which successfully realized the maximum bandwidth of the accelerometer under different external loads.

There are certain errors in the experimental results, especially for the initial structure. For example, when the equivalent inertial acceleration is 1*g*, the initial structure still has a large normalized amplitude at a frequency below 100 Hz, which is inconsistent with the theoretical analysis. The main reason may be that the structure is in a vertical state in the experiment, which means that there is always a 1*g* gravitational acceleration in the direction of the non-sensitive axis as off-axis crosstalk. Thus, when the damping effect is small at low frequencies, there is also a swing in the *x* direction with the vibration along the sensitive axis direction. The Doppler Vibrometer measures the moving speed of the object through the optical path difference between the incident light and reflected light, so this swing may cause a change in the optical path difference, resulting in an error in the measurement. The length of the cantilever beam of the initial structure is longer than that of the optimal structure, so the measurement results of the initial structure are more obviously affected at low frequencies. Nevertheless, the overall errors are in an acceptable range, and the trends of the curves in different acceleration loads show that the bandwidth of the optimal structure is much better than the initial structure, which strongly validates the effectiveness of our optimal scheme.

## 5. Discussion and Conclusions

Dynamic response analysis is critical in MEMS design and application, whereas the current vibration response analyses are mostly limited to small amplitude situations, serving the damping coefficient as a constant. Therefore, large deviations persist when facing large vibration amplitude cases. This paper first extends the vibration response analysis to large vibration amplitude ratios, putting forward a complete model to completely describe the vibration response of the damped vibration system in the attenuation of oscillation and forced vibration. After that, an optimal bandwidth design is proposed based on the optimal design of the sensing unit of a MEMS accelerometer. The optimal design is thoroughly verified by experiment and compared with the initial structure with an extern acceleration load from 1*g* to 10*g*. The experiment results show that the optimal structure can have an excellent bandwidth up to the range from 0 Hz to its natural frequency, much better than the initial structure, proving the effectiveness of the optimal design scheme and laying a solid foundation for further engineering of the accelerometer. For other similar structures of MEMS devices filled with a fluid medium, the optimal design scheme is also applicable and only needs to change the viscous coefficient from air to other corresponding media. The whole vibration response analysis and optimal design scheme can be a sufficient guide and reference for the design of other MEMS devices, carrying significance for the whole community.

## Figures and Tables

**Figure 1 sensors-22-09855-f001:**
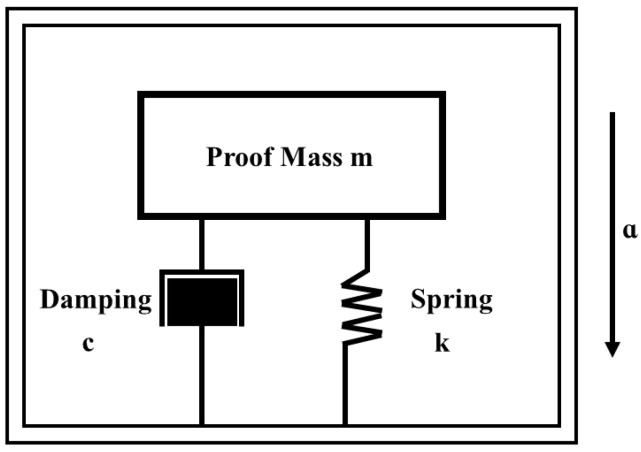
The typical physical model of the sensing unit of the accelerometer.

**Figure 2 sensors-22-09855-f002:**
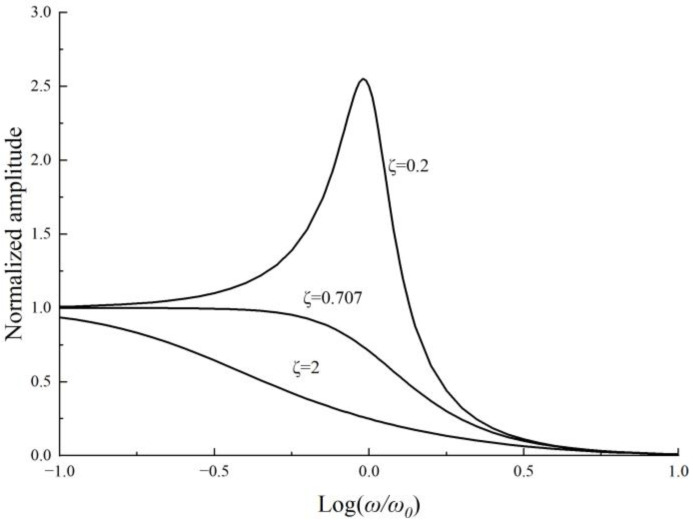
The normalized amplitude response curve in different damping ratios in forced vibration when the amplitude ratio is small.

**Figure 3 sensors-22-09855-f003:**
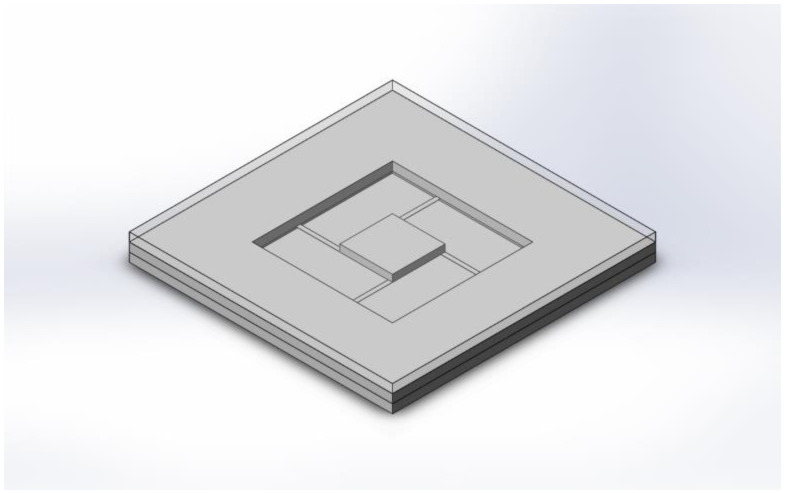
The sensing unit of the accelerometer.

**Figure 4 sensors-22-09855-f004:**
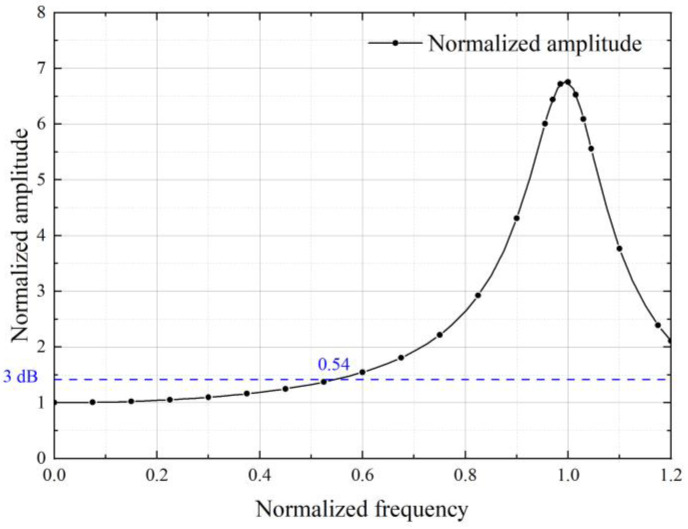
The normalized amplitude response curve when the basic damping ratio is 0.074 in the basic state.

**Figure 5 sensors-22-09855-f005:**
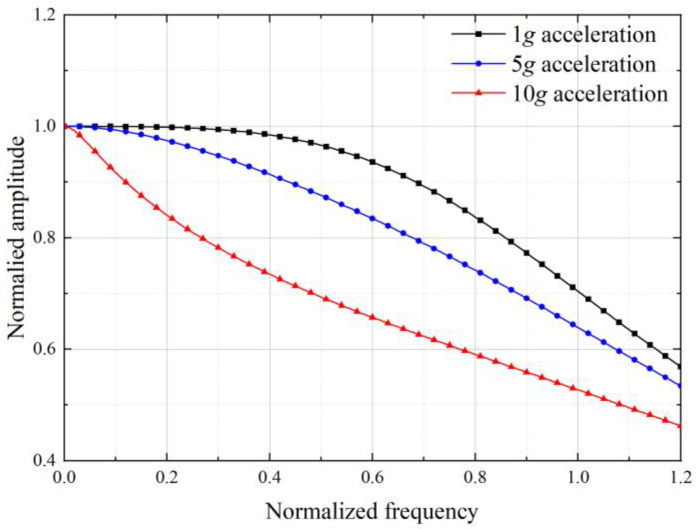
The normalized amplitude response curve in different acceleration load when *ζ*_0_ is 0.707 in the basic state.

**Figure 6 sensors-22-09855-f006:**
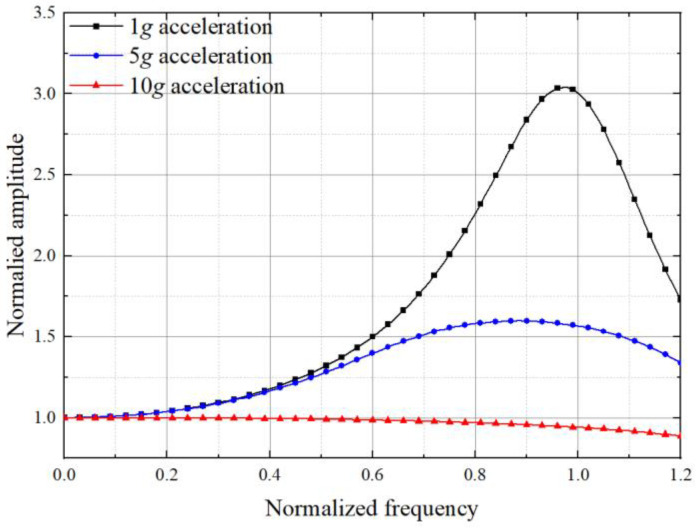
The normalized amplitude response curve in different acceleration load when *ζ_ε_* = 0.707 in large-amplitude state.

**Figure 7 sensors-22-09855-f007:**
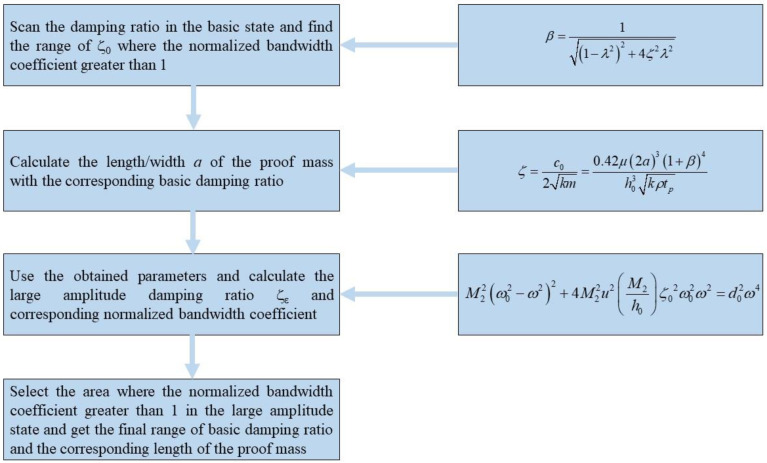
The flowchart of the optimization process.

**Figure 8 sensors-22-09855-f008:**
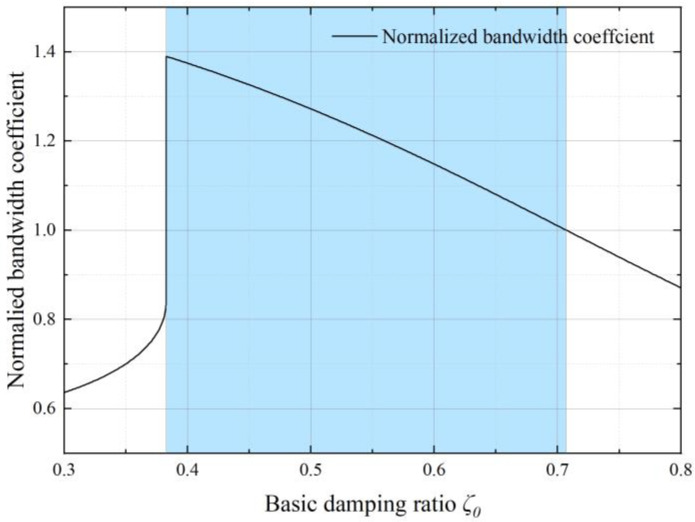
The normalized bandwidth coefficient with different basic damping ratios in the basic state.

**Figure 9 sensors-22-09855-f009:**
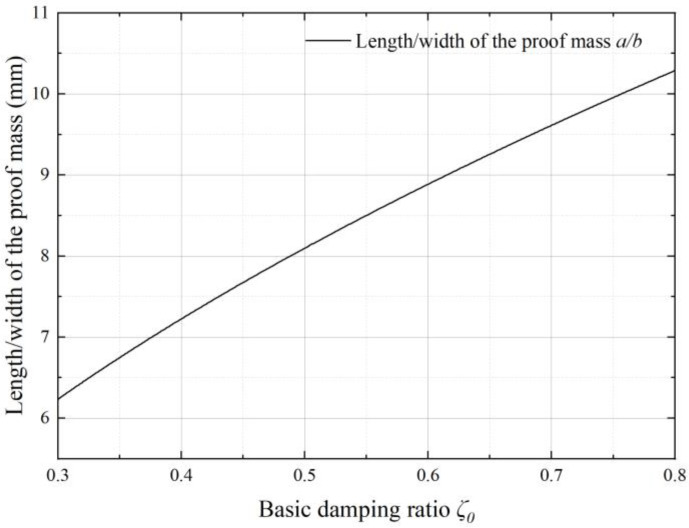
The characteristic length of the proof mass with different basic damping ratios.

**Figure 10 sensors-22-09855-f010:**
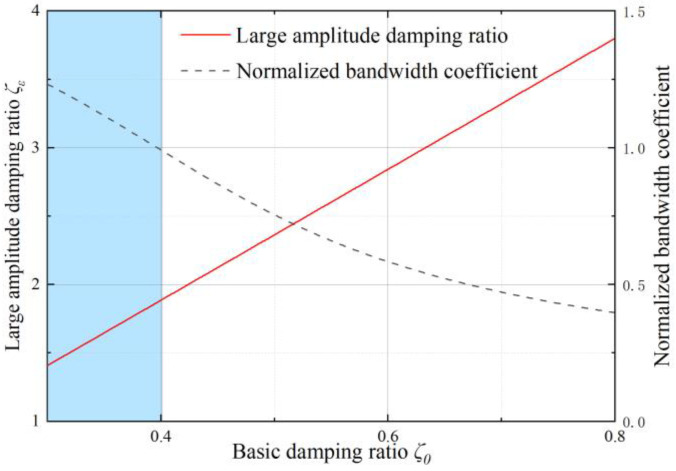
The large-amplitude damping ratio *ζ_ε_* and the normalized bandwidth coefficient with the corresponding basic damping ratio *ζ*_0_

**Figure 11 sensors-22-09855-f011:**
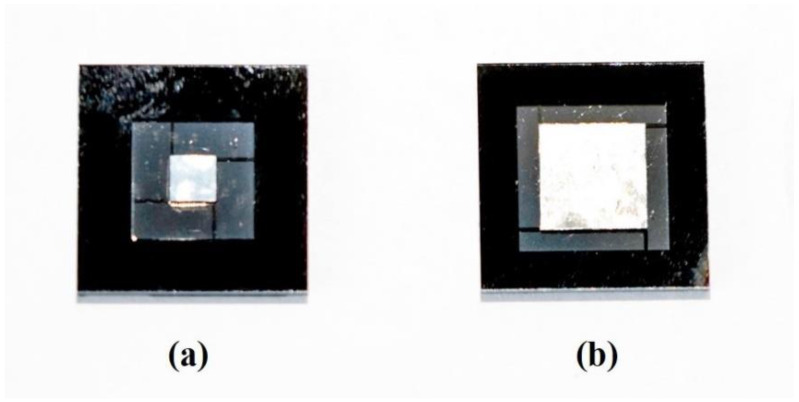
The sandwiched sensing unit. (**a**) The initial structure. (**b**) The optimal structure.

**Figure 12 sensors-22-09855-f012:**
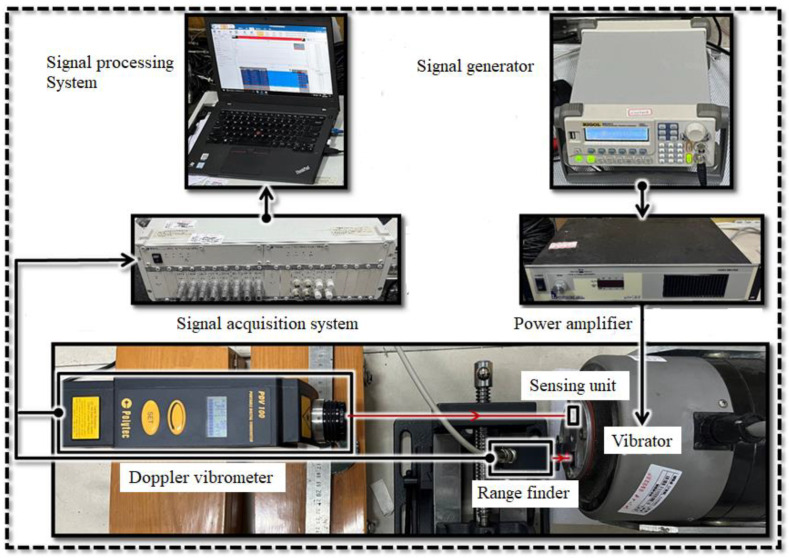
The equipment for the vibration experiment.

**Figure 13 sensors-22-09855-f013:**
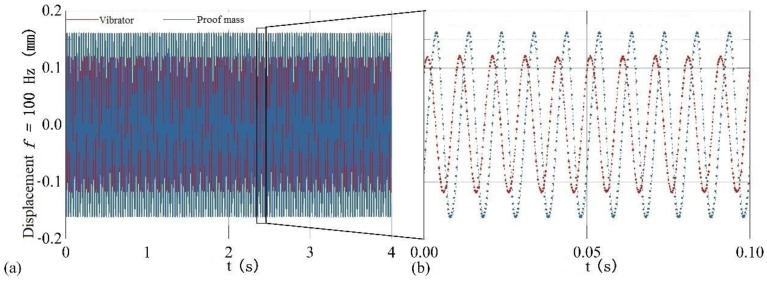
(**a**) The absolute displacement curve of the proof mass and vibrator against time when the equivalent acceleration is 5*g* and the vibration frequency is 100 Hz. (**b**) the sine fitting curves.

**Figure 14 sensors-22-09855-f014:**
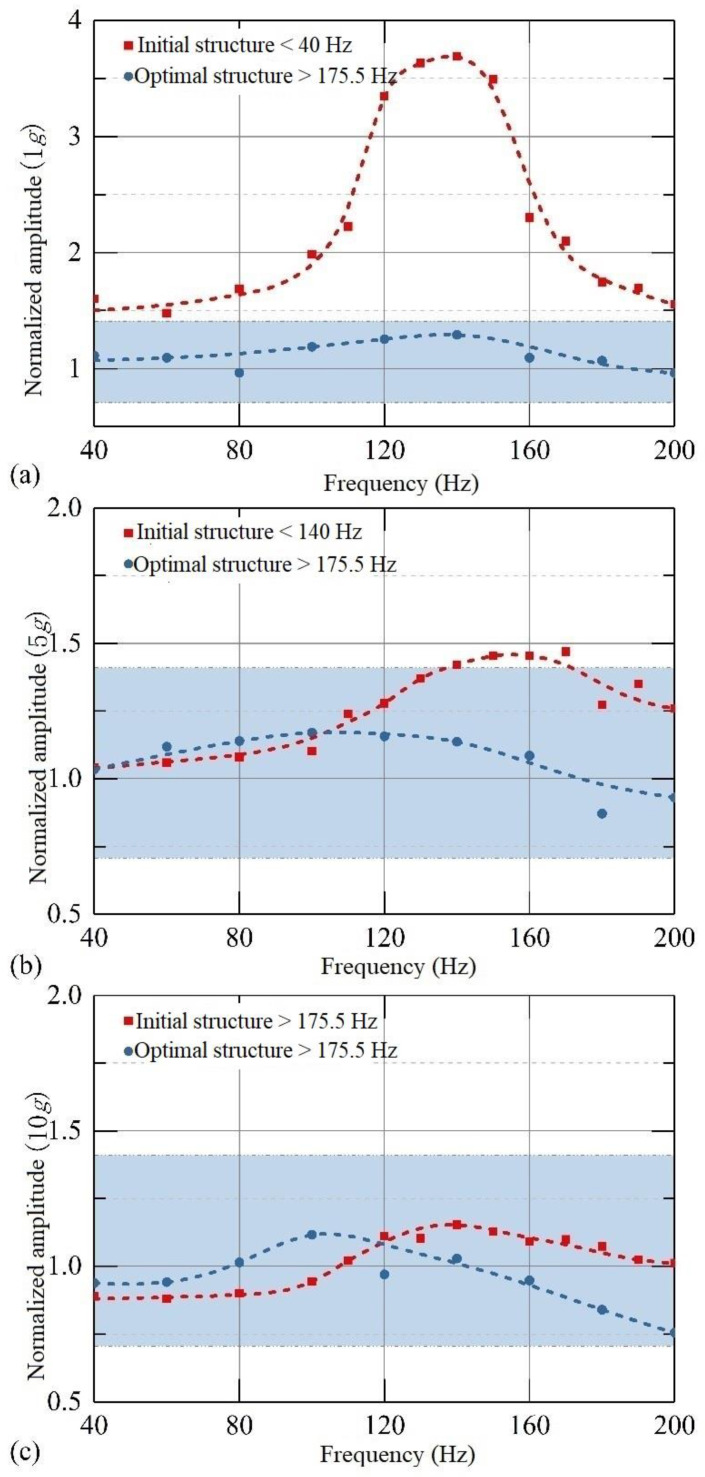
The normalized amplitude response curves in different acceleration loads of the initial structure and the optimal structure. (**a**–**c**) when the equivalent acceleration is 1*g*, 5*g* and 10*g* from top to the bottom.

**Table 1 sensors-22-09855-t001:** The parameters of the initial structure and the optimal structure.

Parameter	Initial Structure	Structure after Optimization
Size of the proof mass	3 mm × 3 mm × 0.4 mm	7.11 mm × 7.11 mm × 0.4 mm
Size of the cantilever beam	2.5 mm × 0.24 mm × 0.01 mm	1.41 mm × 0.24 mm × 0.01 mm
Elastic coefficient of the cantilever beam	10.2 N/m	57.29 N/m
Natural frequency	175.5 Hz	175.5 Hz
Thickness of the air film	100 μm	100 μm
Range of damping ratio	0.074~0.32	0.39~1.76
Bandwidth	0~90 Hz	0~175.5 Hz

## Data Availability

Not applicable.

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
