# Peer review of "Bandwidth Optimization of MEMS Accelerometers in Fluid Medium Environment"

_sensors, 2022, doi:10.3390/s22249855_

Round 1

Reviewer 1 Report

The paper presents the bandwidth optimization of MEMS accelerometers in fluid medium environment.

The article is interesting.

Please:

- correct the article in terms of language,

- determine values of the experiment errors and the results reproducibility. 

Reviewer 2 Report

This manuscript presented a systematic study on the vibration response analysis of the sensing unit of an accelerometer in large amplitude ratios and the optimization scheme to extend the maximum bandwidth. The results can provide a good guide in the damping related analysis and corresponding optimal MEMS design. However, the following comments should be addressed to improve the manuscript.

1. The novelty of this manuscript is not clear. It seems that all the methods and theories are from the references.

2. The specific process of the optimal design was not described clearly. The quality of presentation of this manuscript should be improved. The authors are recommended to add a flow chart for the analysis and optimization.

3. In figure 13, it seems that the advantage of the optimal structure is not obvious.

4. The comparison of the performances of the proposed MEMS accelerometer and the commercial MEMS sensors are recommended to be analyzed.

  •  

Reviewer 3 Report

In this work, the authors gave a vibration response analysis for the sensing unit in the accelerometer by using the analytical method. Further, the sensing unit is optimized to achieve the maximum bandwidth. This work is valuable for the vibration amplitude control of single DOF system. However, there are still some technical problems to answer.

1. In table 1, the authors provided the initial structure parameters and optimized parameters but did not illustrate why select this scheme.  Because this group parameters are not the only one.

2.  the authors should provide the material name and properties of sensing unit, including the proof mass, the cantilever beam.

Round 2

Reviewer 2 Report

This manuscript can be accepted now.